# Na^+^ Lattice Doping Induces Oxygen Vacancies to Achieve High Capacity and Mitigate Voltage Decay of Li-Rich Cathodes

**DOI:** 10.3390/ijms24098035

**Published:** 2023-04-28

**Authors:** Hengrui Qiu, Rui Zhang, Youxiang Zhang

**Affiliations:** College of Chemistry and Molecular Sciences, Wuhan University, Wuhan 430072, China; 2020102030003@whu.edu.cn (H.Q.); zhangrui@whu.edu.cn (R.Z.)

**Keywords:** molten salt template strategy, Na^+^ lattice doped, oxygen vacancy, density functional theory, voltage decay

## Abstract

In this work, we synthesized 1D hollow square rod-shaped MnO_2_, and then obtained Na^+^ lattice doped-oxygen vacancy lithium-rich layered oxide by a simple molten salt template strategy. Different from the traditional synthesis method, the hollow square rod-shaped MnO_2_ in NaCl molten salt provides numerous anchor points for Li, Co, and Ni ions to directly prepare Li_1.2_Ni_0.13_Co_0.13_Mn_0.54_O_2_ on the original morphology. Meanwhile, Na^+^ is also introduced for lattice doping and induces the formation of oxygen vacancy. Therefrom, the modulated sample not only inherits the 1D rod-like morphology but also achieves Na^+^ lattice doping and oxygen vacancy endowment, which facilitates Li^+^ diffusion and improves the structural stability of the material. To this end, transmission electron microscopy, high-angle annular dark-field scanning transmission electron microscopy, X-ray photoelectron spectroscopy, and other characterization are used for analysis. In addition, density functional theory is used to further analyze the influence of oxygen vacancy generation on local transition metal ions, and theoretically explain the mechanism of the electrochemical performance of the samples. Therefore, the modulated sample has a high discharge capacity of 282 mAh g^−1^ and a high capacity retention of 90.02% after 150 cycles. At the same time, the voltage decay per cycle is only 0.0028 V, which is much lower than that of the material (0.0038 V per cycle) prepared without this strategy. In summary, a simple synthesis strategy is proposed, which can realize the morphology control of Li_1.2_Ni_0.13_Co_0.13_Mn_0.54_O_2_, doping of Na^+^ lattice, and inducing the formation of oxygen vacancy, providing a feasible idea for related exploration.

## 1. Introduction

To meet the rapid development of society, lithium-ion batteries (LIBs) have been widely used as efficient and environmentally friendly energy storage devices [1,2,3]. In order to further enhance the application capabilities of LIBs, researchers are trying to find a low-cost, high-capacity, long-cycle cathode material for lithium batteries [4,5]. Therefore, manganese-based lithium-rich materials have gradually entered people’s field of vision. However, due to serious voltage decay and oxygen release, further commercial applications are hindered. The root of the problem can usually be attributed to the following two points: First, the oxygen anion in (Li_1.2_Ni_0.13_Co_0.13_Mn_0.54_O_2_) LLO carries out redox reaction, resulting in oxygen loss in the components and irreversible O_2_ release, which leads to the material structure collapse and battery bulking, bringing more negative effects [6,7]. Another point is that in the highly charged state, the transition metal (TM) ions located at the octahedral positions of the TM layer migrate to the tetrahedral positions of the Li layer. When the three adjacent sites in the Li layer are all vacant, the migration of TM ions becomes easier [8,9]. Therefore, the current research on LLO begins to focus on the modification and optimization of materials.

In general, the redox of anions in LLO is initiated by the extraction of unstable electrons from isolated unhybridized O 2p states on the Li-O-Li configuration, leading to the aggregation of adjacent oxygen atoms to generate O-O dimer, which is consistent with charge compensation. Then the O-O dimer will release O_2_ during the charging-discharging process, which can generate oxygen vacancies (OV) and lead to the collapse of the material structure [10,11]. Although the generation of OV facilitates the diffusion of Li^+^ and increases the capacity of LLO, the disordered structural collapse will seriously deteriorate the electrochemical behavior and safety of the electrode [12,13]. Thereby, it is very important to adjust and optimize the surface oxygen without affecting the overall stability of the material structure. For this reason, inducing OV generation during the preparation of LLO is a promising strategy. At present, alkali metal ions are generally introduced into Li sites to realize the replacement of Li^+^ by heteroions. This strategy is beneficial to the generation of OV, and the previous literature reports have also confirmed that it can effectively improve the electrochemical performance of LLO without affecting the structural stability of LLO [14,15].

This work takes Li_1.2_Ni_0.13_Co_0.13_Mn_0.54_O_2_ as the research object, and we prepared LLO-Na-OV with 1D rod-like structure by a simple molten salt template method. It not only successfully inherits the 1D rod-like structure of the precursor that facilitates charge transport and Li^+^ diffusion, but also introduces Na^+^ during the preparation process to achieve lattice doping and induce the generation of OV. Since Na and Li have similar chemical properties and the radius is relatively large (1.06 A > 0.76 A), the c-axis spacing can be effectively increased after lattice doping, which greatly facilitates the diffusion of Li^+^ during charging-discharging [16]. In addition, characterization and (density functional theory) DFT are used to analyze the influence of OV generation on local TM ions, and its effect on electrochemical properties is explained by mechanism. Therefore, compared with the Pristine-LLO sample, the optimized LLO-Na-OV achieves superiority in initial Coulombic efficiency, discharge capacity, median voltage, cycle stability, and other aspects, which also confirms the potential value of this synthesis strategy for improving the electrochemical performance of LLO.

## 2. Results and Discussions

Figure 1 shows the synthesis process of the 1D rod-like structure LLO-Na-OV prepared by the molten salt template method. We first prepared hollow MnO_2_ with a rod-like structure by a simple hydrothermal method and then mixed it with nickel salt, cobalt salt, and lithium salt to a homogeneous state by stoichiometric number. Simultaneously, sodium chloride was added four times the number of moles expected to be synthesized. At a high temperature of 900 °C, NaCl would form a molten salt, which could fully contact the raw materials of each component and avoid the drawbacks of particle agglomeration in the traditional solid-state method. Moreover, hollow MnO_2_, as a manganese source and a synthetic template, provided numerous anchor points for Li, Co, and Ni ions due to its unique structure, so that LLO could be directly prepared on the original morphology. During the synthesis process, the materials were all in the NaCl flux environment, which also provided convenience for LLO to do Na^+^ lattice doping and induced oxygen vacancies. In addition, ICP detection was adopted to verify that the stoichiometry of the as-prepared samples was consistent with the preset target Li_1.2_Ni_0.13_Co_0.13_Mn_0.54_O_2_, and the detailed parameters are shown in Appendix A.

The XRD patterns of Pristine-LLO, 10-LLO-Na-OV, 15-LLO-Na-OV, and 20-LLO-Na-OV are shown in Figure 2. It can be seen that all samples have two strong peaks (003) and (104), corresponding to the rhombohedral phase structure of LiMO_2_ (M=Ni, Co, Mn). Moreover, the XRD pattern is in good agreement with the structure of hexagonal α-NaFeO_2_ (PDF#87-1564, R$\bar{3}$m space group) except for some weak diffraction peaks of Li_2_MnO_3_ between 20–25° that represent C2/m monoclinic symmetry [17,18]. Furthermore, the (006)/(012) peaks and (018)/(110) peaks at around 38° and 65° show well splitting, which proves that the samples have a well-layered structure. Interestingly, the c-axis in the crystal structure affects the positions of the (003) and (104) diffraction peaks. It can be seen from the XRD magnification pattern of Figure 2 that with the increase of Na^+^ doping amount, the (003) and (104) diffraction peaks also shift to the left [19]. This phenomenon is because Na^+^ with a larger ionic radius (1.02 A > 0.76 A) is doped into the Li layer, resulting in the increase of the c-axis plane spacing, which is conducive to the diffusion of lithium ions in the lattice and plays a crucial role in improving the electrochemical performance of the sample [20]. In addition, the XRD patterns of LLO—850 °C and LLO—950 °C are also shown in Appendix A, similar to 15-LLO-Na-OV, both of them have the same characteristic peaks and R$\bar{3}$m space group. However, because they are the products of pre-condition exploration and the electrochemical performance is not satisfactory (Appendix A), the micromorphologies are also given in Appendix A, which will not be discussed below.

Subsequently, the lattice doping of Na^+^ was further demonstrated by Rietveld refinement, preferably choosing the R$\bar{3}$m space group of typical layered α-NaFeO_2_ (PDF#87-1564) as the basis of calculation, as shown in Appendix A. In addition, the refinement parameters of the samples are also given (Appendix A). By analyzing the reliability factors Rp and Rwp, it can be seen that they are both less than 10%, indicating that the structure model has a high degree of agreement with the sample structure [21]. Meanwhile, it can be observed that the c/a values of all samples are greater than 4.9, implying that they have a well-layered structure [22]. It is worth mentioning that the c-axis lengths of 10-LLO-Na-OV, 15-LLO-Na-OV, and 20-LLO-Na-OV are larger than those of the Pristine-LLO, which indicated that Na^+^ is a successfully introduced lattice and this behavior increases the interlayer distance of Li^+^ plates [23].

The micromorphologies of the samples were observed by SEM. It can be seen that the MnO_2_ obtained by the hydrothermal method possesses a hollow 1D rod-like structure with a length and radius of about 1 μm and 50 nm, respectively (Appendix A). Then, LLO is directly prepared by using it as a template to provide a large number of anchor points for Li, Co, and Ni ions in NaCl flux, which perfectly inherit the 1D rod-like structure of MnO_2_ (Appendix A). However, different reaction times did not produce significant changes in the morphology of the samples, and they all maintained a 1D rod-like structure (Appendix A). In contrast, the LLO obtained without the molten salt template method (Pristine-LLO) is irregular and agglomerated (Appendix A), indicating that MnO_2_ as a template is destroyed during the preparation process, making it difficult to impart to LLO. At the same time, MnO_2_ particles with damaged morphology will produce agglomeration at high temperatures, thus further affecting the morphology of LLO. The above analysis shows that the unique hollow structure not only facilitates Li^+^/Ni^2+^/Co^2+^ to contact the inner and outer surfaces of MnO_2_ but also provides enough space for the crystal growth of LLO. At the same time, it also effectively maintains the 1D rod-like, avoiding the same agglomeration phenomenon as Pristine-LLO.

TEM, HRTEM, and HAADF-STEM were used to further collect the structural information and observe the morphology of 15-LLO-Na-OV, as shown in Figure 3. Intuitively, 15-LLO-Na-OV has a well-crystalline structure, and the 1D rod-like morphology of MnO_2_ is effectively inherited (Figure 3a). In HRTEM images, two groups of lattice spacing d1 = 0.42 nm (Figure 3b) and d2 = 0.47 nm (Figure 3c) can be measured, corresponding to (020) crystal plane of Li_2_MnO_3_ monocline structure and (003) crystal plane of layered α-NaFeO_2_ structure, respectively [24,25]. In addition, it can be observed that at the edge of the sample, small ridges (red dotted lines) appear in the lattice of the (003) crystal plane, which may be formed by the Na^+^ substitution of the Li site in the Li layer. Furthermore, significant stacking faults due to the increase in lattice spacing after Na^+^ entry into the lattice are also observed (green dashed line). Then, the atomic resolution HAADF-STEM images further confirm the Na^+^ lattice doping in LLO (Figure 3d), and it can be intuitively seen that the highly crystallized layered structure along the (003) crystal plane in the sample and the interlayer substitution of Na^+^ in the Li plate. Therefore, through the above analysis, it can be demonstrated that 15-LLO-Na-OV achieves Na^+^ doping substitution of the Li site in the Li layer during the preparation process, which is also because the Li site in the Li layer is more mobile and easier to be replaced than Li^+^ in TM site [26]. In addition, the EDS mapping of 15-LLO-Na-OV is also given (Figure 3e), indicating the uniform distribution of Ni, Co, Mn, Na, and O atoms in the sample.

In order to analyze the valence state of the elements in the sample and the influence of Na^+^ doping, XPS measurement technology was applied. As shown in Appendix A, the peaks of Ni, Co, Mn, O, and Na elements can be found in all samples, except that the Na element cannot be found in the Pristine-LLO sample spectrum. For the Ni 2p spectrum (Figure 4a), Ni 2p3/2, Ni 2p1/2, and their satellite peaks are distributed around 854, 872, 861, and 879 eV, respectively. Moreover, two fitting peaks of Ni^3+^ and Ni^2+^ at 856 and 855 eV can be obtained by peak fitting of Ni 2p3/2, which is consistent with the literature [27]. While in the Co 2p spectrum (Figure 4b), Co 2p3/2 and Co 2p1/2 are located at 780 eV and 795 eV, respectively. Of course, fitting the Co 2p3/2 spectrum peaks also yielded Co^3+^ and Co^2+^ fitting peaks at 779 eV and 781 eV [28]. Similarly, for the Mn 2p spectrum, Figure 4c shows two peaks at 642 and 654 eV, representing Mn 2p3/2 and Mn 2p1/2. After fitting to Mn 2p3/2, two fitting peaks at 642 and 643 eV point to Mn^3+^ and Mn^4+^ [29]. Through careful comparison, it can be seen that in all samples, the valence state of Ni element (Ni^2+^/Ni^3+^ both remain around 1.81) remains unchanged. On the contrary, the valence ratio of Mn and Co has shifted. From Pristine-LLO to 20-LLO-Na-OV, Mn^4+^/Mn^3+^ are 2.06, 1.94, 1.85, and 1.70, respectively. Co^3+^/Co^2+^ are 1.72, 1.69, 1.66, and 1.62, respectively. This can be attributed to the presence of OV which reduces the valence state of Mn and Co elements. For this purpose, the spectrum of O 1s is presented in Figure 4d. After fitting, three strong peaks at 529, 531, and 533 eV can be obtained, corresponding to lattice oxygen atoms, oxygen vacancies, and surface-adsorbed oxide ions in the sample, respectively [30]. Intuitively, the OV peaks in the Pristine-LLO samples are significantly lower than those in the LLO-Na-OV samples prepared by the molten salt template method, indicating that they generate numerous OVs during the preparation process [31]. Interestingly, the more OV in the sample and the fewer oxide ions adsorbed on the surface, the more it can suppress the oxygen loss of the Li_2_MnO_3_ component during the charging-discharging process, and it can also facilitate the diffusion of Li^+^ [32]. This will be shown later by the lithium-ion diffusion coefficient (Appendix A).

DFT calculations were introduced to explain the influence between OV and the valence states of Co/Mn elements and to elucidate the electrochemical performance mechanism. To simplify the calculation, two simple models were established: Li_19_Ni_2_Co_2_Mn_9_O_32_ and Li_18_Na_1_Ni_2_Co_2_Mn_9_O_31_, representing Pristine-LLO and LLO-Na-OV samples, respectively (the associated crystal structure diagram of the models are shown in Appendix A). The total density of states and the predicted density of states (TDOS/PDOS) of the two samples are shown in Figure 5a,b. In general, the spin-up part occupies the electronic state near the Fermi level, and the spin-down part has a band gap. This is similar to some of the literature reports [33]. The valence and conduction bands in the DOS diagram mainly contain TM 3d states and O 2p states, while their coincidence on the PDOS implies that the TMO_6_ octahedral unit mediates strong TM-O interactions. When LLO introduces Na^+^ and induces oxygen vacancies, the DOS curve does not change significantly, but the band gap widens slightly (from 0.35 eV to 0.44 eV). The LLO-Na-OV sample has a lower unoccupied state peak, which can reduce the probability of internal charge transitions and is beneficial to the stability of the structure [34]. In addition, average charge density curves are plotted to illustrate the effects of Na^+^ lattice doping and OV on the samples (Figure 5c,d). The average charge density of the Li layer increases due to the substitution of Li^+^ by Na^+^ doping in the Li layer but has no other effects. After the OV is generated, the average charge density of the O layer will decrease, thereby reducing the attraction to nearby Co and Mn elements. Therefore, the valence states of Co and Mn elements in the LLO-Na-OV sample will be lower than those in the Pristine-LLO sample, which is also consistent with the XPS analysis. In-depth, the corresponding deformation charge density, and its 2D section images are also displayed (Figure 5e), visually presenting the different degrees of influence of the O element on the electron gain and loss of the surrounding TM elements. It can be seen that the O atoms and TM atoms exhibit red and blue colors, respectively. The blueness of the Mn and Co atoms near OV decreases with the decrease of the redness of the O atom, which means that the Mn and Co atoms lose electrons to a lesser degree, thus reflecting their lower valence. However, the difference is that the blueness of the Ni atoms in the image has not changed, indicating that its valence state has not changed due to the generation of OV. This conclusion is highly consistent with the average charge density curve and XPS analysis results.

In order to test the LLO-Na-OV obtained by the molten salt template method and evaluate the improvement of its electrochemical performance, a coin-type half-cell was assembled with a voltage range of 2.0–4.8 V. Subsequently, CV tests were performed on all samples at a scan rate of 0.1 mV s^−1^ to analyze their electrochemical behavior, as shown in Figure 6. The oxidation peak at about 4.03 V in the CV curve corresponds to the slope region of 2.0 V to 4.45 V during the initial charging curve, representing the oxidation process of Ni^2+^ and Co^3+^ into Ni^4+^ and Co^4+^ [35,36]. At the same time, Li^+^ is also deintercalated from the LiMn_0.33_Ni_0.33_Co_0.33_O_2_ component to form octahedral vacancies. The oxidation peak at 4.61 V corresponds to the slope region of 4.45 V to 4.8 V during the initial charging curve, representing the deintercalation of Li^+^ and the oxidation of O^2−^ in the Li_2_MnO_3_ component (Li_2_MnO_3_ → Li_2_O + MnO_2_ and O^2−^/O^−^ or O^2−^/O_2_) [37]. This conclusion is confirmed by the double plateau of the curve in Figure 7a. In addition, three reduction peaks corresponding to the reduction processes of Co^4+^/Co^3+^, Ni^4+^/Ni^2+,^ and Li^+^ embedded activated MnO_2_-rich components are observed at 4.42, 3.72, and 3.23 V [38]. However, as the CV test proceeds, the oxidation peak at 4.61 V disappears in the second/third cycle because the electrochemical behavior of the Li_2_MnO_3_ component in LLO is irreversible [39].

Figure 7a shows the initial charge-discharge curves of different samples at 0.1 C. In the charging stage, two gentle slopes are shown, corresponding to the oxidation of Ni^2+^ and Co^3+^ and the irreversible deintercalation of Li^+^ to form lithium oxide. This phenomenon is also highly consistent with CV analysis. In addition, compared to Pristine-LLO, the discharge capacity and initial Coulombic efficiency of LLO obtained by the molten salt template method are leapfrogged (detailed data are shown in Appendix A). This improvement may be due to the existence of OV and the appropriate amount of Na^+^ lattice doping, which not only effectively reduces the irreversible lithium oxide in the sample, but also increases the lattice spacing to facilitate the intercalation/deintercalation of Li^+^, which promotes the charge transfer kinetics of Li^+^ [40]. The rate performance is shown in Figure 7b, under different current rates, 15-LLO-Na-OV has the highest rate performance, while 10-LLO-Na-OV and 20-LLO-Na-OV are next, which further illustrates the importance of proper Na^+^ doping.

The median voltage is introduced as an important parameter to verify that Na^+^ lattice doping and generation of oxygen vacancies can effectively reduce the voltage decay of LLO. Figure 7c presents the plot of median voltage versus cycle period for all samples. It can be seen that the initial median voltages of 10-LLO-Na-OV, 15-LLO-Na-OV, and 20-LLO-Na-OV samples are 3.25, 3.24, and 3.28 V respectively, and there are still high voltages of 2.73, 2.82 and 2.74 V after 150 cycles. However, the initial median voltage of Pristine-LLO is only 3.07 V, which is as low as 2.50 V after 150 cycles. Analysis of the data shows that the average decay voltages per cycle of 10-LLO-Na-OV, 15-LLO-Na-OV, and 20-LLO-Na-OV are 0.0034, 0.0028 and 0.0036 V, respectively, which are lower than the average attenuation voltage per cycle of 0.0038 V of the Pristine-LLO. This result strongly proves that the Na^+^ doping and OV introduced by the molten salt template synthesis strategy can make LLO obtain a higher median voltage, and at the same time significantly alleviate its voltage attenuation, which shows the hope for the practical application of the material. It is worth mentioning that cycling performance is also critical for LLO (Figure 7d). The capacity retention rate of 15-LLO-Na-OV is as high as 90.02% after 150 cycles at 1 C, while 10-LLO-Na-OV and 20-LLO-Na-OV are 87.66% and 70.03%, respectively. It shows that proper Na^+^ doping also plays a certain role in the stability of the material structure. In contrast, Pristine-LLO exhibited only 56.40% capacity retention after 150 cycles. This result can be attributed to the following two aspects [41,42]: On the one hand, Na^+^ doping occupies part of Li^+^ sites in the Li plate, and the Na^+^ in the charging-discharging process will not be intercalated/desorbed in the lattice as Li^+^, which greatly ensures the stability of the original layered structure and lays a solid foundation for the high performance of LLO. On the other hand, the presence of OV also plays a pinning effect, which provides the resistance of LLO during phase transition, so that the material can effectively maintain the layered structure. Therefore, the electrochemical performance of 15-LLO-Na-OV is improved compared with some of the literature (Appendix A) [43,44,45,46,47].

In order to further verify the effect of Na^+^ introduction on Li^+^ diffusion kinetics, the Nyquist and equivalent circuit diagrams of the samples are shown in Appendix A. The Nyquist curves are composed of a slanted line in the low-frequency region and a semicircle in the intermediate-frequency region, representing the diffusion behavior of Li^+^ in the electrode material (Warburg impedance) and the charge transfer impedance (Rct) at the electrode/electrolyte interface, respectively. The intercept between the curve and the real axis in the high-frequency region is the film resistance (Rs) of Li^+^ diffusing through the surface CEI film [48,49]. Therefore, in order to study the Li^+^ diffusion kinetics of the sample, the Li^+^ diffusion coefficient (D_Li_^+^) is given by the following equation [50]:(1)$$D_{Li}{}^{+}=\frac{R^{2}T^{2}}{2A^{2}n^{4}F^{4}C^{2}\sigma^{2}}$$

Here, each letter in the formula is as follows: R and T are the gas constant and absolute temperature, A and n represent the effective working area of the electrode and the number of electrons transferred by each molecule, while F and C represent the Faraday constant and Li^+^ molar concentration, respectively. Furthermore, σ can be obtained by relating Z’ to ω^−1/2^ (Appendix A), corresponding to the Warburg factor, given by the following equation [51]:(2)$$Z\prime =R_{s}+R_{ct}+\sigma \omega^{-1/2}$$

In summary, Appendix A records the Rs and Rct values and corresponding D_Li_^+^ for all samples. After comparison, it can be seen intuitively that compared with the Pristine-LLO, the Rs and Rct of the LLO-Na-OV series before and after cycling are both smaller, suggesting that Na^+^ doping improves the diffusion kinetics of Li^+^. At the same time, the D_Li_^+^ of the preferred 15-LLO-Na-OV is even several times higher than Pristine-LLO, which further confirms this conclusion. The Rs and Rct of all samples increased after cycling due to the existence of side reactions. In addition, Appendix A also shows the XRD pattern and SEM image of 15-LLO-Na-OV after cycling, which has no significant change compared with that before cycling (Figure 2 and Appendix A), providing strong evidence for the superior structural stability of the material itself.

Raman spectra can be obtained through the normal vibration frequency of unique molecular and crystal information and then used for the analysis of micro-domain phase structure. By fitting and comparing the Raman spectra of samples before and after cycling, it can be seen that there are several prominent Raman bands at 594, 472, 416, and 341 cm^−1^ (Figure 8). The Raman peaks at 594 and 476 cm^−1^ correspond to M-O symmetric stretching (A_1g_) and O-M-O symmetric deformation (E_g_) in the R$\bar{3}$m symmetric Raman active vibration mode, respectively, while the Raman peaks at 416 and 341 cm^−1^ correspond to Li_2_MnO_3_-like structures resulting from the reduced local symmetry of C2/m. This is in good agreement with theoretical predictions for hexagonal (R$\bar{3}$m) and monoclinic (C2/m) crystals previously reported in the literature [52]. It is also confirmed by the XRD pattern. Interestingly, the E_g_ vibrations of 10-LLO-Na-OV, 15-LLO-Na-OV, and 20-LLO-Na-OV are shifted to the right relative to Pristine-LLO, probably due to the induction of OV during the preparation process (detailed data are shown in Appendix A). In addition, the spinel structure in the sample is also fitted. By analyzing the I_R_/I_S_ value, we can intuitively observe the quality of the layered structure of the sample (I_R_ is the pink curve representing the layered structure, and I_S_ corresponds to the spinel structure, marked as the purple curve). At the beginning of the cycle, the I_R_/I_S_ of Pristine-LLO, 10-LLO-Na-OV, 15-LLO-Na-OV, and 20-LLO-Na-OV are 2.48, 2.34, 2.28, and 2.31, respectively, meaning that they all had well-layered structure. The introduction of Na^+^ lattice doping and induction of OV generation will also produce certain stacking faults (Figure 3c), but this significantly improves the electrochemical performance of LLO. After 150 cycles, the I_R_/I_S_ of Pristine-LLO, 10-LLO-Na-OV, 15-LLO-Na-OV, and 20-LLO-Na-OV decreased to 0.46, 1.05, 1.03, and 0.99, respectively. Therefore, their ΔI_R_/I_S_ are 2.02, 1.29, 1.25, and 1.32, respectively. This shows that the Pristine-LLO has a large number of layered structures transformed into spinel phases after long-term repeated charge-discharge processes, while the LLO-Na-OV samples obtained by the molten salt template strategy can effectively strengthen the structural stability of the cathode material particles [53]. It can be concluded that the anti-deintercalation of Na^+^ lattice doping plays a supporting role for the layered structure during the charging-discharging process, and the OV pinning effect increases the phase transition barrier. Their synergistic effect also plays a crucial role in maintaining the stability of the lamellar structure. This result also affirms the analysis of cycle performance. Moreover, the FWHM of I_R_ and I_S_ increases after cycling (Appendix A), implying that the degree of amorphization increases with the progress of the electrochemical reaction [54,55]. This series of changes will lead to structural degradation and then electrochemical performance attenuation [56]. Therefore, based on the above analysis, it is proved that this preparation strategy has important guiding significance and potential value for optimizing the electrochemical performance of LLO.

## 3. Materials and Methods

### 3.1. Materials and Chemical Reagents

The reagents used in this experiment, such as nickel nitrate hexahydrate, cobalt nitrate hexahydrate, lithium hydroxide monohydrate, potassium permanganate, hydrochloric acid, sodium chloride, and ethanol were purchased from Sinopharm Chemical Reagent Co., Ltd. (Shanghai, China), and the purity was analytically pure without further purification.

### 3.2. Synthetic Process

Firstly, 0.608 g of KMnO_4_ was dissolved in 70 mL of distilled water and stirred to a homogeneous solution, then 1.30 mL of concentrated HCl (37%, wt%) was dropped and stirred again for 30 min. After that, the solution was transferred to a 100 mL polytetrafluoroethylene stainless steel autoclave and reacted at 140 °C for 12 h. The brown precipitate collected after cooling was washed several times until the pH of the filtrate reached 7, and then dried overnight at 100 °C for use. Since then, rod-shaped MnO_2_ was obtained.

Subsequently, Ni(NO_3_)_2_·6H_2_O, Co(NO_3_)_2_·6H_2_O, MnO_2_ and LiOH·H_2_O (excess 5%, mol) with stoichiometric ratios were mixed with NaCl (the molar amount of NaCl was controlled to be the expected synthesis four times the number of moles of LLO), and dispersed in ethanol with sealing and stirring for 4 h. Then, the ethanol was slowly evaporated and the resulting mixture was ground and transferred to a muffle furnace for calcination at 900 °C for 15 h (heating rate of 5 °C min^−1^). After cooling, the sinter was taken out for grinding, rinsed several times to remove the NaCl flux, and dried overnight at 100 °C to obtain LLO with Na^+^ doped induced oxygen vacancies. Furthermore, in order to optimize the electrochemical performance of LLO, a series of explorations at different temperatures (calcination at 850 and 950 °C for 15 h) and at different times (calcination at 900 °C for 10, 15 and 20 h) were carried out. For convenience, the prepared samples were labeled as LLO-850 °C, LLO-950 °C and 10-LLO-Na-OV, 15-LLO-Na-OV, 20-LLO-Na-OV, respectively. Moreover, LLO samples obtained without NaCl molten salt method were named Pristine-LLO.

### 3.3. Material Characterization

The microscopic morphology and element distribution of the samples were given by scanning electron microscopy (SEM, JEOL JSM-7100F) and transmission electron microscopy (TEM, JEOL JEM-F200), and further atomic resolution images were obtained by high-angle annular dark-field scanning transmission electron microscopy (HAADF-STEM, JEM-ARM200F). The crystal structure and phase changes were determined by X-ray diffractometer (XRD, D8 Discover) and micro-Raman spectroscopy (Raman, XperRam S instruments). In order to ensure that the molecular formula of the synthesized samples met the expected target, inductively coupled plasma atomic emission spectrometry (ICP, Optima 7300 DV) was used to determine the element content. The valence information and presence of elements were characterized by X-ray photoelectron spectroscopy (XPS, ESCALAB250Xi). Moreover, for the SEM and XRD characterization of the samples after cycling, the cells were disassembled in a glove box to take out the electrode sheets, washed with dimethyl carbonate (DMC), and then moved to a vacuum drying box to dry for use.

### 3.4. Electrochemical Measurements

To obtain the cathode electrode, the active material, acetylene black and polyvinylidene fluoride (PVDF) binder were dispersed in N-methyl-2-pyrrolidone (NMP) at 8:1.5:0.5 to form a slurry, which was then uniformly coated over the aluminum foil collector. Subsequently, the CR2016 button battery was assembled in a glove box in an argon atmosphere using lithium metal sheet as anode, 1 M LiPF_6_ as electrolyte (ethyl carbonate/dimethyl carbonate, volume ratio, 1:1) and Celgard 2300 microporous film as the diaphragm. The rate capacities and long cycling were measured on Neware CT-4008T battery testing system with a voltage range of 2.0–4.8 V and 1 C = 250 mA g^−1^. Cyclic voltammetry (CV) and electrochemical impedance spectroscopy (EIS) were performed on a CHI760C electrochemical workstation. The electrochemical behavior of the electrodes was analyzed by CV under the conditions of operating voltages ranging from 2.0 to 4.8 V and scan rates of 1.0 mV s^−1^, while the impedance of each material was recorded by EIS in the frequency range of 100 kHz to 0.01 Hz. 

It can be seen that they all inherit the rod-like shape of MnO_2_, and the overall difference is not significant. However, LLO-950 °C appears to be more agglomerated, and some irregular squares appear, which may be due to sintering caused by high temperature.

## 4. Conclusions

In a word, the Na^+^ lattice doping of LLO and the induced generation of OV are realized by a simple molten salt template method. This method widens the distance between Li^+^ plates in the preparation process, facilitates the embedding/de-embedding of Li^+^, and promotes the charge transfer reaction kinetics. At the same time, the induced OV inhibits the oxygen loss of the Li_2_MnO_3_ component during the charging-discharging process, which provides a boost to alleviate the LLO voltage decay. In addition, their synergistic effect also greatly ensures the structural stability of the material, thereby achieving the goal of long-cycle high-capacity retention. In particular, the 15-LLO-Na-OV sample not only has an initial capacity as high as 282 mAh g^−1^ but also has a capacity retention rate of 90% after 150 cycles. Moreover, the voltage decay is also alleviated, and the voltage decay per cycle is only 0.0028 V. Therefore, we can optimistically believe that this strategy provides a feasible idea for the realization of LLO with high capacity-low voltage attenuation, and lays a solid foundation for the practical application of LLO.

## Figures and Tables

**Figure 1 ijms-24-08035-f001:**
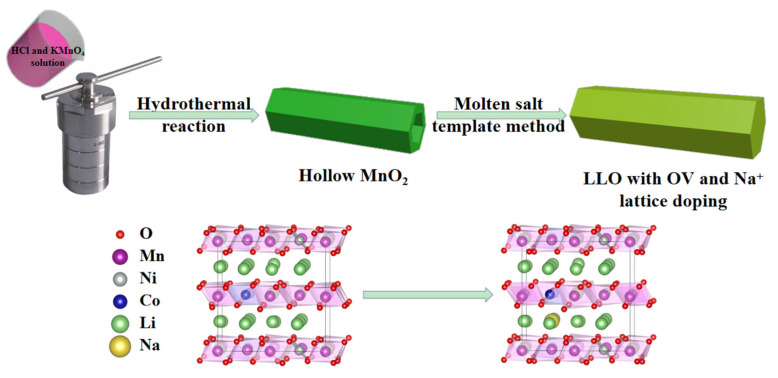
The illustration of the molten salt template strategy.

**Figure 2 ijms-24-08035-f002:**
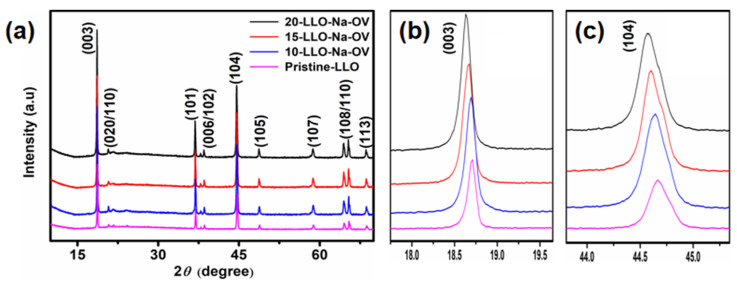
(**a**) XRD patterns of Pristine-LLO, 10-LLO-Na-OV, 15-LLO-Na-OV, and 20-LLO-Na-OV; (**b**,**c**) the magnification pattern of (003) and (104) peaks.

**Figure 3 ijms-24-08035-f003:**
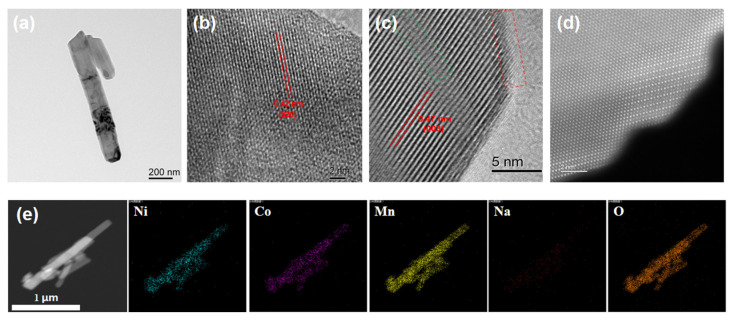
(**a**) TEM, (**b**,**c**) HRTEM, (**d**) HAADF-STEM images, and (**e**) EDS mapping of 15-LLO-Na-OV.

**Figure 4 ijms-24-08035-f004:**
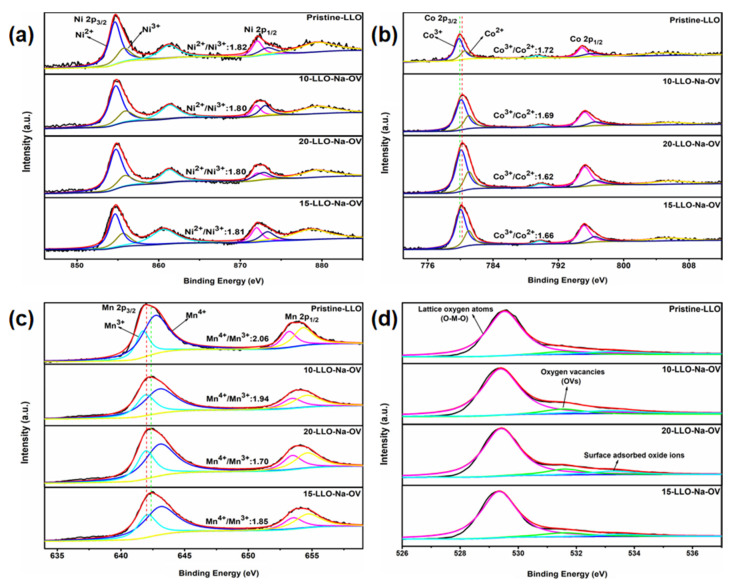
XPS spectra of (**a**) Ni 2p, (**b**) Co 2p, (**c**) Mn 2p, and (**d**) O 1s for all samples.

**Figure 5 ijms-24-08035-f005:**
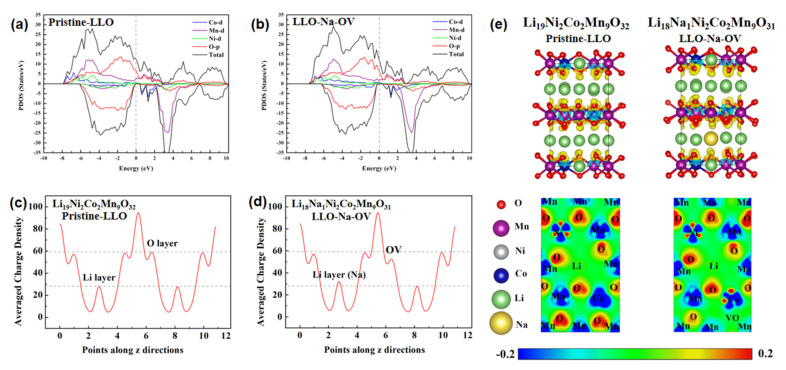
(**a**,**b**) Total/projected density of states, (**c**,**d**) average charge density curves, and (**e**) deformation charge density and its 2D section images of Pristine-LLO and LLO-Na-OV models.

**Figure 6 ijms-24-08035-f006:**
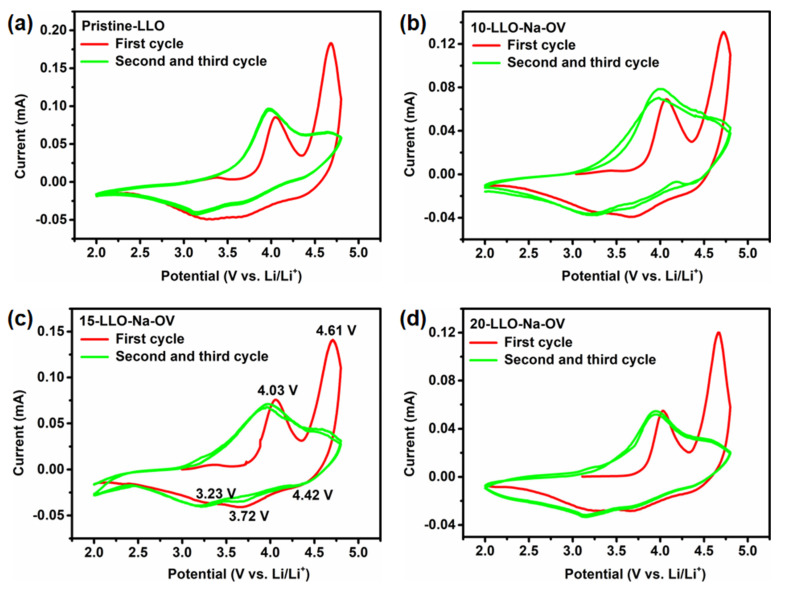
The CV plots at a scan rate of 0.1 mV s^−1^ of (**a**) Pristine-LLO, (**b**) 10-LLO-Na-OV, (**c**) 15-LLO-Na-OV, and (**d**) 20-LLO-Na-OV.

**Figure 7 ijms-24-08035-f007:**
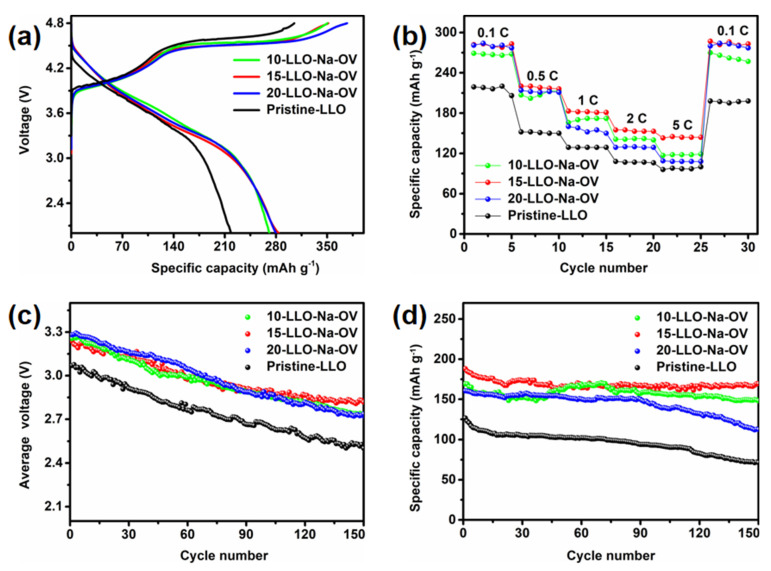
The electrochemical properties of Pristine-LLO, 10-LLO-Na-OV, 15-LLO-Na-OV, and 20-LLO-Na-OV: (**a**) initial charge-discharge curves at 0.1 C, (**b**) rate performances at 0.1–5 C, (**c**) median voltage curves at 1 C, and (**d**) cycling performance at 1 C.

**Figure 8 ijms-24-08035-f008:**
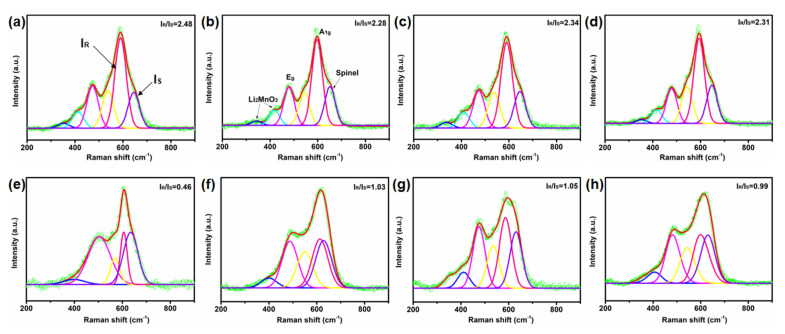
Raman spectra of (**a**,**e**) Pristine-LLO, (**b**,**f**) 15-LLO-Na-OV, (**c**,**g**) 10-LLO-Na-OV and (**d**,**h**) 20-LLO-Na-OV: (**a**–**d**) before and (**e**–**h**) after cycling.

## Data Availability

The data presented in this study are available from the corresponding author upon request.

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
