# Peer review of "Na+ Lattice Doping Induces Oxygen Vacancies to Achieve High Capacity and Mitigate Voltage Decay of Li-Rich Cathodes"

_ijms, 2023, doi:10.3390/ijms24098035_

Round 1

Reviewer 1 Report

The Authors are recommended to characterize the present sample in more detail.

(1)    The composition of the present sample should be described.

(2)    The Authors are recommended to analyze the XRD profile by, for example, Rietveld analysis, to confirm the consistency of the estimated composition and the XRD profile.

Reviewer 2 Report

The authors submitted a manuscript with an interesting idea, to investigate by Raman post-mortem samples. The manuscript needs extensive editing before being accepted. It is challenging to add comments in a document without line numbering.

1. Abstract text should not contain acronyms.

2. Define the acronyms in the introduction before using them.  

3. "Moreover , hollow MnO 2 , as a manganese source and a synthetic template, provide d numerous anchor points for Li, Co, and Ni ions due to its unique structure, so that LLO could be directly prepared on the original morphology. " - please elaborate on the unique structure of MnO2 that allows the insertion of anchor points.

4. In the same paragraph, what is "ICP detection"?

5.  .The above analysis shows that NaCl presents a molten state at 900 ℃, which provides a reaction environment that is conducive to the full contact of various components, and the hollow MnO 2 as a source of manganese and a template provides.... - this phrase needs rephrasing.

6. Why is there no Na in Fig 3. e but there is Na in XPS? Please comment. 

7. Why did the authors show TEM/HRTEM/HAADF-STEM analysis only for one sample?

8. "On the contrary, the valence ratio of Mn and Co has shifted. From Pristine LLO ..." - are these ratios reproducible? is the ratio computed from one scan of one sample?

9. The caption of Fig 6 needs additional details. The scan rate should be in the Figure caption. It appears that there are more CVs on the same graph. It would be easier if the authors color the cycles and explain it in the Figure caption.

10. Regarding Fig. 8., can you add a Table with the deconvolution details? Are the peak positions identical before and after cycling? How about the FWHM? It appears that the Ir and Is peaks have a larger FWHM after cycling. Please comment.

Round 2

Reviewer 1 Report

The manuscript was revised well according to the Reviewers' comments, and now it is acceptable for publication in the journal.

Reviewer 2 Report

Accept as is.